# Methods to Address Self-Selection and Reverse Causation in Studies of Neighborhood Environments and Brain Health

**DOI:** 10.3390/ijerph18126484

**Published:** 2021-06-16

**Authors:** Lilah M. Besser, Willa D. Brenowitz, Oanh L. Meyer, Serena Hoermann, John Renne

**Affiliations:** 1Department of Urban and Regional Planning, Institute for Human Health and Disease Intervention (I-HEALTH), Florida Atlantic University, Boca Raton, FL 33431, USA; 2Departments of Psychiatry and Behavioral Sciences and Epidemiology & Biostatistics, University of California San Francisco, San Francisco, CA 94121, USA; Willa.Brenowitz@ucsf.edu; 3Department of Neurology, University of California Davis, Sacramento, CA 95817, USA; olmeyer@ucdavis.edu; 4Center for Urban and Environmental Solutions (CUES), Department of Urban and Regional Planning, Florida Atlantic University, Boca Raton, FL 33431, USA; shoermann@fau.edu (S.H.); jrenne@fau.edu (J.R.)

**Keywords:** epidemiological methods, causality, reverse causation, self-selection, bias, neighborhood, built environment, brain health, Alzheimer disease, cognition

## Abstract

Preliminary evidence suggests that neighborhood environments, such as socioeconomic disadvantage, pedestrian and physical activity infrastructure, and availability of neighborhood destinations (e.g., parks), may be associated with late-life cognitive functioning and risk of Alzheimer’s disease and related disorders (ADRD). The supposition is that these neighborhood characteristics are associated with factors such as mental health, environmental exposures, health behaviors, and social determinants of health that in turn promote or diminish cognitive reserve and resilience in later life. However, observed associations may be biased by self-selection or reverse causation, such as when individuals with better cognition move to denser neighborhoods because they prefer many destinations within walking distance of home, or when individuals with deteriorating health choose residences offering health services in neighborhoods in rural or suburban areas (e.g., assisted living). Research on neighborhood environments and ADRD has typically focused on late-life brain health outcomes, which makes it difficult to disentangle true associations from associations that result from reverse causality. In this paper, we review study designs and methods to help reduce bias due to reverse causality and self-selection, while drawing attention to the unique aspects of these approaches when conducting research on neighborhoods and brain aging.

## 1. Introduction

Studies on the influence of neighborhood environments (NE) (i.e., social and built environments (BE)) on brain health are still in their infancy but are growing rapidly and provide tentative evidence that our community environments may affect the brain throughout the lifespan [1,2,3]. Greater neighborhood socioeconomic disadvantage has been associated with worse baseline cognition [4,5], greater decline in cognition over time [6], and total and regional brain volumes from magnetic resonance imaging (MRI) [7] among older adults. Neighborhood racial/ethnic segregation has been linked to poorer cognitive health outcomes in middle and older aged individuals [8,9,10]. In addition, measures of the BE such as greater land use mix (e.g., mix of retail and residential) [11], access to retail destinations [12], public transportation availability [13], greater walkability [14] (i.e., environment conducive to walking by providing multiple destinations and density of street connections), and greater greenness/park space access [15,16,17,18] have been associated with various measures of brain health in older adults including diagnoses of Alzheimer’s disease and related disorders (ADRD). This body of research is typically rooted in the socioecological framework that posits that beyond individual level determinants of brain health, including age and genetics, there are likely higher-level social determinants of health (SDOH) operating, including neighborhood and community environments. SDOH affect environmental exposures and access to support, resources, and opportunities that ultimately affect a population’s morbidity and mortality.

Yet, even the most rigorous studies of the NE and brain health to date, such as those that employ population-based cohorts with longitudinal follow-up or natural experiments, can still be biased due to attrition (e.g., individuals with the outcome of interest dropping out at a higher rate), non-representativeness of the sample compared to the general population, and competing causes/residual confounding. Table 1 presents some of the methodological challenges unique to studying NEs and brain health, such as defining the neighborhood geographic boundary, capturing the neighborhood exposure, defining the neighborhood construct, typical reliance on studies of older adults, and the lag between development of pathology for ADRD and the diagnosis of dementia. We first present these broader issues to set the context for our more detailed focus on self-selection and reverse causation. Reverse causation occurs when the outcome precedes and results in the exposure (Figure 1a). Cross-sectional studies are prone to this potential issue because temporal ordering of exposure and outcome cannot be established. This may also be an issue for studies on outcomes with long preclinical or subclinical periods. Reverse causation in which changes in brain health predict the neighborhood one resides in is an example of self-selection. However, self-selection can also bias NE-brain health studies not through reverse causation, but through confounding, in which the lifestyle or neighborhood preferences influence residential choices (Figure 1b). Any association between the NE and brain health may instead be attributable to lifestyle behaviors and preferences that preceded neighborhood choice and that are also associated with the brain health outcome.

Studies of brain health including ADRD risk are particularly susceptible to reverse causality bias because older adults are likely to move to new neighborhoods following the development of cognitive or physical impairment including dementia [19]. Thus, any resulting association between a particular NE and cognition or dementia risk may be due to these late life residential moves. In addition, ADRD neuropathology and subtler cognitive changes can occur a decade or more prior to a diagnosis of ADRD, further complicating matters. In these cases, researchers may presume that associations between NEs and ADRD risk were free of bias from neighborhood self-selection because residential environments were measured prior to disease development. However, ADRD may more subtly affect brain health and functioning up to two decades prior to a dementia diagnosis [20,21]. Sometimes, more than a year can occur between full-blown dementia symptoms and the receipt of a diagnosis [22]. Thus, residential moves in mid to late life may be in part due to these subtler and longer-running changes in brain health.

This paper aims to review methods for addressing reverse causation and self-selection in the context of the broader challenges of research on NEs and brain health. We provide a cursory overview of available methods to address reverse causation and self-selection, example studies that have employed these methods, and two case studies. The goal of the paper is not to provide specific instructions on how to conduct the outlined methods, but to provide a summary of the methods with useful references and considerations to guide future research on NEs and brain health.

## 2. Methods to Address Reverse Causality and Self-Selection

The goal of research is typically to identify causal effects (e.g., what NE factors improve brain health and what factors hasten cognitive decline). These can then inform interventions or policy changes to improve wellbeing. However, often we deal with statistical associations and findings from observational studies that may be impacted by a number of biases including self-selection and reverse causation. Based on the idea of counterfactual models, i.e., that there are unobserved potential outcomes for each observed outcome, the goal of causal inference would be to a compare observed to counterfactual outcomes among the same individuals. Since this is not possible, we must substitute other individuals to approximate the counterfactual outcome. If the individuals in the two groups are not similar along other parameters, then this will introduce bias. The goal of many research strategies is to make exposed/unexposed groups similar along potentially biasing factors.

In this section, we describe various approaches that can be used to enhance causal inference in studies of NEs and brain health, such as propensity score weighting and natural experiments (Table 2). Since few studies of the NE and brain health have incorporated these methods, we provide examples from the published literature on other health outcomes (e.g., physical activity).

### 2.1. Randomized Control Trial/Experiment

Randomized control trials (RCT) are considered the gold standard study design for estimating causal effects [26]. RCTs employ randomization to help balance potential confounders across intervention groups and also establish clear temporal order of cause and effect. An example of a neighborhood-based RCT was the Moving to Opportunity trial in which individuals from high poverty neighborhoods in public housing were chosen randomly to move into either neighborhoods of high or low poverty. In one published study of the participants, families and children in that study were interviewed three years after their moves [27]. Those who moved to neighborhoods of low poverty experienced less distress (parents) and anxiety/depressive symptoms (boys). However, RCTs may not be feasible for understanding neighborhood effects due to their costs, the ethical implications, and the difficulty of recruiting participants (or neighborhoods) for interventions. If conducted well, RCTs have low chance of bias by self-selection and reverse causation, but use of RCTs has been extremely rare in NE and health research due to the hurdles to implementation. Thus, we would not propose the use of a RCT given the difficulties for NE and health studies, but instead present it as the gold standard method with which other methods are generally compared.

### 2.2. Multivariable Regression—Covariate Adjustment

The most common approach to address self-selection is through covariate adjustment in multivariable regression models [28,29]. Participant or neighborhood characteristics that are known or theorized to influence neighborhood choice are included as additional predictors in regression models. Such variables could include demographics (e.g., age, education, marital status, race/ethnicity) or measures that describe or relate to choice of neighborhood (e.g., preference for urban vs. rural or ranking of importance of neighborhood characteristics) [29]. For example, a study of individuals from eight neighborhoods in Northern California examined cross-sectional associations between neighborhood BE characteristics and frequency of walking to the store, while controlling for self-reported neighborhood preferences when determining where to live (i.e., accessibility, safety, physical activity options, socializing, outdoor spaciousness, and attractiveness) [30]. The authors found that BE characteristics including distance to destinations were significantly associated with walking to the store even after controlling for neighborhood self-selection characteristics.

Covariate adjustment in regression models may help capture key confounders, but rests on the assumption that confounders are accurately measured and included in the models. Capturing participant preferences and factors that accurately determine self-selection into neighborhoods may be challenging. As noted previously, measuring neighborhood preferences may not capture all facets of self-selection into neighborhoods because individuals may have difficulty articulating and prioritizing their reasons and preferences [31]. Furthermore, adjustment in regression models does not easily account for potential reverse causality of brain health influencing neighborhood choice.

### 2.3. Multivariable Regression—Propensity Scores/Inverse Probability Weighting

Propensity scores employ a two-step process to control for confounding. First, probability (propensity) for exposure is modelled in a first stage and scores derived from predicted probabilities are created to include in the second stage, or primary analytic model. Various techniques for propensity scores have been developed, some of which include matching based on propensity score, and others the exclusion of outlier propensity scores. Several studies on the NE have used propensity scores [32,33], as one advantage of this technique is that many covariates can be included when developing the propensity scores, and yet power will not be sacrificed in the full models. This may allow for better control of confounding by self-selection but still relies on similar assumptions as a traditional multivariable regression model.

Another analytic strategy to improve causal inference is to use weighting techniques based on propensity scores to account for potential biases. In some cases, adjustment may lead to overadjustment (e.g., if potential confounders are also potential mediators). In other cases, regression adjustment may lead to bias, such as adjusting for variables that are common effects of both NE and brain health (or their confounders) (aka collider-stratification or selection bias), or in cases of effect modification of confounders [34,35]. Therefore, weighting techniques can be used to remove potential sources of bias not amenable to covariate adjustment techniques. Inverse probability weighting (IPW), inverse probability of treatment weights (IPTW), and marginal structural models use a two-stage process to first estimate the effect of important confounders on probability of treatment/exposure and then incorporate the inverse predicted probabilities into analytic models [36,37]. Weights can be defined to account for time-varying confounding [37], selection bias (e.g., modelling probability of selection instead of treatment) [38], and missing data [39]. Weights are used to calculate whether observations are either over-represented or under-represented when compared to a target population with no differences along potential confounders or compared to the original sample in the case of addressing selection/missing data. Observations that are under-represented are given increased weights (up-weighted) while observations that are over-represented are given decreased weights (e.g., down-weighted). This results in a pseudo-population balanced along potential confounder or selection variables [36]. While useful in the case when confounding and selection processes are measured, this approach cannot account for unmeasured factors.

### 2.4. Longitudinal Study Design

Longitudinal study design is an often recommended method to reduce the chances of self-selection and reverse causation bias [29]. Presumably, multi-year follow-up of individuals allows for the estimation of the effect of an exposure/treatment early on in the follow-up and longitudinal change in outcome or later development of disease (i.e., establishes temporality of exposure and outcome). Methods such as within-person analyses can account for time-invariant confounders that are measured and unmeasured [29]. However, longitudinal designs alone do not fully address potential biases by self-selection and reverse causation because the previously mentioned complications of studies of NEs and brain health, namely the impact of neighborhood preferences that can change over time (i.e., time-varying) and the potential for preclinical and prodromal (undiagnosed brain disease) conditions to affect neighborhood choice/moves and thus the neighborhood characteristics under study.

### 2.5. Restricted/Stratified Sample

Restriction or stratification is another traditional method to account for potential confounding by self-selection or reverse causality. Studies can restrict analytic samples along some measurement (e.g., stated preference for location) or by limiting to a small area or certain neighborhoods. For instance, a study of 20–65 year olds in Belgium used a pre-existing questionnaire [40] to assess neighborhood selection factors (i.e., house price, preference for city center, quietness, social/emotional reasons, walkability) [41]. The authors then performed a sub-analysis to compare associations between an objective measure of neighborhood walkability and several outcome measures including walking for transportation in the full sample versus the restricted sample that reported high importance of neighborhood walkability characteristics. The associations the authors observed using the full sample remained significant and similar in the restricted sample.

Restriction/stratification may also be used to attempt to account for reverse causation. Individuals with cognitive impairment, low cognitive tests scores, or other biologic measures could be excluded to attempt to eliminate participants already experiencing brain changes that could affect neighborhood choice/characteristics. In longitudinal settings, this can also be applied such that cases of dementia within a certain period of baseline neighborhood measurements are excluded. The difficulty remains in that prodromal disease (e.g., undiagnosed of Alzheimer’s disease) can last many years and could thus still impact residential moves/neighborhood characteristics in the years leading up to diagnosis.

### 2.6. Quasi-Experiments and Natural Experiments

Quasi-experimental and other causal inference approaches have become increasingly popular as methods that use observational data to approximate randomized designs [42]. When RCTs are not feasible, there may be “natural experiments” that occur which can be leveraged to estimate causal effects. A natural experiment is when an intervention has occurred but the circumstances the lead to the intervention were not controlled by the researchers [43]. Some examples are specific policy changes or laws that affect the NE, development projects in neighborhoods, natural events, or other quasi-experimental factors. Natural experiments are posited to not be influenced by participant characteristics, which can help eliminate bias from unmeasured or mismeasured confounders. Natural experiments also establish temporal order of events. Estimates from natural experiments are thus not as susceptible to self-selection or reverse causality and may provide an unbiased estimate of the causal effects of NE on brain health. Various analytic approaches can be used in natural experiments, such as interrupted time series, pre-post designs, instrumental variable analyses [44], regression discontinuity analyses [42,45], and difference-in-difference approaches [46].

Example natural experiments in the NE and health literature include studies focused on new BE amenities, such as rail lines, parks, or supermarkets [47,48,49,50]. While some of these studies suggest that BE additions have had positive impacts on health, the evidence has been mixed on whether changes to the BE have had positive effects on health outcomes such as physical activity, diet, and obesity [50]. The limitations of these types of natural BE experiments are as follows. (1) The individuals studied before and after the BE change may be different (e.g., measuring change in health outcomes at census tract level does not account for movers). (2) A change to the BE is not the same as a change in exposure to that BE characteristic (e.g., a new supermarket or rail line is not necessarily used by the individuals) [51]. (3) Competing factors associated with the change in health are not necessarily captured (e.g., new neighborhood gyms). (4) The change in health outcome needs to be measured on the appropriate time scale (e.g., detection of changes in brain health requires a longer follow-up than changes in physical activity). (5) Population-based samples remain difficult to obtain. (6) It is difficult to find neighborhoods that are good controls/comparison groups (i.e., differ from intervention comparison in no important way related to the health outcome other than the intervention itself). Further, (7) environment changes are often known after the fact, and it is challenging to collect pre-change health data for a pre-post study [29].

Given these limitations, some researchers have suggested a reorientation of these natural experiments to focus on a comparison of individuals who move versus those who do not move, to examine how the change in BE due to moves affects health outcomes [51]. For example, one study examined changes in body mass index and physical activity levels following residential relocation among participants in the Multi-Ethnic Study of Atherosclerosis [52]. The authors found that individuals who moved to more walkable neighborhoods (i.e., 10-point higher Walk Score) compared to their prior residence increased transport walking by 16.04 min per week (95% confidence interval (CI) = 5.13, 29.96).

Despite positive findings such as the above example, a systematic review found that the majority of natural experiment studies of the impact of the BE on physical activity had moderate to high risk of bias (e.g., did not control for important confounders, inadequate control sites) [53]. In addition, the previously observed impacts of a residential move to health outcomes such as physical activity may not be readily observed in studies of brain health outcomes. One can imagine a comparison of movers and non-movers could be confounded by unmeasured characteristics that affect mobility and residential choice and thus confound the associations. Regardless of the type of natural experiment (change of BE due to additions/renovations or moves), the expected impact of a BE change on cognition or ADRD incidence may not be large and immediate enough to rule out competing causes.

#### 2.6.1. Pre-Post Designs, Difference-In-Difference

One common quasi-experimental approach is to leverage longitudinal data and examine before and after health effects resulting from neighborhood changes, reducing concerns for reverse causality. These studies typically rely on individual data to compare before and after effects. Studies have employed pre-post analyses to examine changes in health due to natural experiments such as the construction of light rail [47]. However, these types of studies may still be susceptible to self-selection bias and not all studies have identified a control/comparison group [29], which limits causal inference.

Difference-in-difference (DID) is another approach that can capitalize on the use of longitudinal data. DID is a quasi-experimental method to estimate the effect of a specific intervention or NE/BE change by comparing the difference in outcomes over time between a population that received the intervention/exposure and a population that did not (control) [43]. For instance, DID has been used to estimate the association between neighborhood condition and weight changes while accounting for potential self-selection due to movers and non-movers [54]. The method has also been used to estimate the effect of neighborhood investment on physical activity and body mass index adjusting for general changes over time [55]. In addition, a DID approach has been used to examine self-selection [56], by estimating how baseline health may relate to changes in NE characteristics after the baseline time period. DID is useful when longitudinal data are available but rests on the assumption that the time-varying trends in the treated/exposed group are parallel to the control group in the absence of the intervention [43]. Ultimately, there still may be potential bias, such as that due to unmeasured time-varying confounding factors.

#### 2.6.2. Instrumental Variables

Instrumental variable analyses [44,57] have emerged as a popular approach to estimate causal effects of exposures/treatments. Assumptions that an instrumental variable must meet are: 1. The instrument must be associated with the outcome. 2. The instrument must affect the outcome only through the exposure. 3. The instrument is not associated with confounders of exposure and outcome. 4. Estimated effects of instruments are the same across differing levels of the instrument (monotonicity). Instrumental variables are expected to be randomly determined and thus in theory allow for an unbiased estimate of the exposure and outcome association.

The use of instrumental variables has been infrequent thus far in neighborhood environment and health studies [58,59,60,61,62]. Example instruments in these previous studies of NEs and diet and obesity included: (a) distance to arterial roads and non-residential zones, (b) distance to nearest highway, and (c) buildable land available for fast-food restaurants within a half-mile of the participant’s residence. These instruments were chosen to try to allow for causal inference between neighborhood characteristics and health outcomes independent of self-selection factors. To our knowledge, instruments have not been employed in studies of NEs and brain health. Thus, we explore this method to investigate reverse causality in the second case study presented in Section 3.

Overall, natural experiment designs rest on additional assumptions that may be untestable, such as the assumption that the neighborhood intervention/instrumental variable is independent of factors that also influence brain health (e.g., no confounding). These study designs are still susceptible to other sources of bias such as selection bias (e.g., due to selective survival) and competing risk of death. Despite the outlined difficulties of conducting natural experiments, if designed well, they may provide the best hope at causal inference among the discussed methods. Incorporating multiple approaches that are not as susceptible to the same biases can help lead to triangulation of evidence. Comparing natural/quasi-experimental approaches to regression estimates may help understand the potential biasing effects of unmeasured attributes of self-selection and reverse causation.

## 3. Case Studies

In this section, we provide two case studies that illustrate the use of four methods described above: (1) adjustment for self-selection as a covariate; (2) restriction/stratification by the self-selection variable; (3) instrumental variable analysis to account for self-selection; and (4) inverse probability weighting to study reverse causation.

### 3.1. Case Study 1: Accounting for Self-Selection via Adjustment and Propensity Scores

#### 3.1.1. Sample

For our first case study, we evaluate the influence of neighborhood self-selection. In 2020 and 2021, 151 residents in South Florida (FL), US, consented to and completed a one-time Community Quality of Life Survey administered by Florida Atlantic University investigators. The survey of five South FL neighborhoods (Abacoa and the Heights of Jupiter in Jupiter, FL, Mirasol in Palm Beach Gardens, FL, and Historic Neighborhoods of West Palm Beach, FL, USA) included questions on the respondents’ demographics (age, sex, have kids in home, race/ethnicity, income, education, marital status), neighborhoods and communities (e.g., perceptions, access to amenities), travel, employment, quality of life, satisfaction, sense of community, lifestyle, and health. The study was approved by the Florida Atlantic University Institutional Review Board. The goal of these analyses was to determine whether methods to account for self-selection result in different associations between living in the Abacoa neighborhood and self-reported minutes of walking and bicycling per week.

#### 3.1.2. Exposure

The exposure was residing in the neighborhood of Abacoa in Jupiter, FL (n = 42) compared to the other South Florida neighborhoods surveyed (n = 109). Abacoa is a 2055-acre master planned, mixed-use community based on the concepts and principles of traditional neighborhood development (TND) [63].

#### 3.1.3. Outcomes

The outcome measures were (1) self-reported minutes of walking in the neighborhood per week (sample mean: 131.2; SD: 303.5), and (2) self-reported minutes of bicycling per week (sample mean: 37.3; SD: 79.4).

#### 3.1.4. Self-Selection Measure

The neighborhood preference/self-selection measure was based on a question on whether nearby neighborhood amenities (e.g., parks, nearby restaurants and entertainment) was the primary reason respondents chose their homes (yes: n = 57; no: n = 94). The other choices for primary reason were affordability, easy commute, family and friends nearby, in a historic district, home and yard design, and quality schools.

#### 3.1.5. Methods

We conducted linear regression analyses to examine the association between living in the Abacoa neighborhood (versus the other surveyed neighborhoods) and minutes of walking and bicycling per week. All models controlled for age, sex, marital status, income, education, race/ethnicity, having children at home, employment status, reported always driving to destinations, and participant appraisals of neighborhood availability of parks/open space and ability to walk to shops/dining (agree or strongly agree versus somewhat agree, disagree, strongly disagree).

Three analyses were conducted to try to account for neighborhood self-selection:Method 1: Stratified the regression analyses by (1) those who reported the primary reason for choosing the current home was the nearby neighborhood amenities and (b) those who did not report the primary reason for choosing the current home for was the nearby neighborhood amenities.Method 2: Controlled for primary reported reason for choosing current home was nearby neighborhood amenities (yes versus no).Method 3: Calculated propensity scores in multivariable logistic regression that represented the respondent’s probability of living in the Abacoa neighborhood based on the variables collected in the survey that were hypothesized to be related to living in Abacoa and to minutes of walking/bicycling per week. The variables included in the models were age, sex, race/ethnicity, education, income, kids at home, marital/partner status, employment status, always drive to destinations, number of household vehicles, exercise for 30 min at least 5 times a week, self-reported physical health status (good/excellent versus neutral/fair/poor), and primary reason chose home was nearby neighborhood amenities. The propensity scores were then converted to stabilized inverse probability weights that were applied to the multivariable linear regression models.

All analyses were conducted in SAS v9.4 using PROC LOGISTIC and PROC REG, and weights were applied using the ‘weight’ statement.

#### 3.1.6. Results

The participant characteristics are presented in Table 3. The majority of respondents were between the ages of 40 and 69 (73%), were women (60%), were white (85%), and had family incomes of ≥$100,000 (63%). After stratifying the associations by the neighborhood amenities self-selection variable (Method 1), we observed higher estimates for the associations between living in Abacoa and minutes of walking and bicycling per week amongst those who chose their homes for the nearby neighborhood amenities (walking estimate: 227.2, 95% CI: −65.4, 419.8; bicycling estimate: 61.5, 95% CI: −2.5, 125.5) compared to those who did not move based on neighborhood amenities (walking estimate: 55.2, 95% CI: −1.71, 112.2; bicycling estimate: 19.5, 95% CI: −11.6, 50.6) (Table 4). However, none of the stratified estimates were statistically significant at *p* < 0.05, likely due to the reduced sample sizes in the stratified groups. After controlling for neighborhood amenity self-selection as a covariate (Method 2), we found that living in Abacoa was associated with a higher reported number of minutes of neighborhood walking and cycling per week (walking estimate: 128.8, 95% CI: 15.5, 242.1; bicycling estimate: 31.6; 95% CI: 2.6, 60.7) when compared to survey respondents from the other neighborhoods (Table 4).

The multivariable model that did not control for self-selection and did not employ inverse probability weights demonstrated associations between living in Abacoa and more walking and bicycling per week (walking estimate: 132.1, 95% CI: 19.4, 244.8; bicycling estimate: 33.8, 95% CI: 4.6, 63.0) (Table 5). These estimates were similar to the estimates from Method 2 except slightly higher. However, when accounting for self-selection using inverse probability weights (Method 3), the estimate for the association between living in Abacoa and minutes walking/week was reduced but still statistically significant, and the estimate for the association with minutes bicycling per week was attenuated and no longer statistically significant (walking estimate: 96.7, 95% CI: 2.1, 191.3; bicycling estimate: 24.2, 95% CI: −2.6, 51.0).

#### 3.1.7. Case Study 1 Conclusions

In these analyses, we demonstrate that inverse probability weighting may be preferable to adjustment or stratification when attempting to account for neighborhood self-selection. However, a potential disadvantage is that inverse probability weights usually increase confidence intervals due to lower study power [64]. In particular, restricting by the self-selection variable is likely to reduce the power to detect a significant association and would only be potentially useful with reasonably large sample sizes. While inverse probability weighting attenuated the findings when compared to adjustment or no adjustment for self-selection as a covariate, the association between living in Abacoa and walking/week remained and suggests that either neighborhood self-selection is either only a partial confounder or that the measure of self-selection used in our study did not fully capture the pertinent construct of self-selection. Altogether, the findings differed enough by method to provide caution to investigators studying NEs and health, particularly for studies using cross-sectional data. Finally, an important caveat that must be noted is the lack of a racially/ethnically diverse sample, which could bias the findings. Thus, this case study was not meant to demonstrate causal associations, but instead to illustrate methods that could be employed to address self-selection bias.

### 3.2. Case Study 2: Instrumental Variable Analysis to Study Reverse Causation

#### 3.2.1. Sample

In our second case study, we evaluate the possibility of reverse causation. We use an instrumental variable to study the association between episodic memory as the predictor and neighborhood greenness as the outcome, modeled after a similar method employed in prior studies [65,66,67]. Data came from 243 participants from the University of California, Davis Alzheimer’s Disease Research Center (ADRC) in Northern California. Enrollment and follow-up methods have been described in detail elsewhere [68,69]. Briefly, participants were recruited into the ADRC through two routes: (1) memory clinic referrals and (2) community outreach. Participants received multidisciplinary diagnostic evaluations that followed the same protocol and included a detailed medical history and a physical and neurological exam.

We obtained data on the participants’ age, sex, race/ethnicity (White, Black, Latino, Other), education (years), genetic risk for Alzheimer’s disease (i.e., presence of one or more apolipoprotein E (APOE) ε4 alleles), recruitment source (clinic versus community), site (Bay area versus Sacramento, California), episodic memory scores, and US Census tract (residential location). The participants included in our sample were on average 76 years old (standard deviation = 7.1), 57% were female, 25% were Black, 23% were Latino, 46% were White, 6% were other race/ethnicity, and their mean years of education was 13.6 (SD = 4.2). Participants came from 202 US Census tracts. The study was approved by the Florida Atlantic University (FAU) Institutional Review Board (IRB) (University of California Davis IRB relied on FAU IRB).

#### 3.2.2. Normalized Difference Vegetation Index

Neighborhood normalized difference vegetation index (NDVI) [70] was calculated in ArcGIS using LANDSAT satellite imagery for each participant’s US Census tract residential location in 2010. NDVI is measured on a scale of −1 to +1 (more positive = greener/healthier vegetation) and is based on visible and near infrared light reflectance of vegetation. Our measure of interest was mean NDVI value for the participants’ Census tract.

#### 3.2.3. Episodic Memory

Participants’ cognition was measured using the Spanish and English Neuropsychological Assessment Scales (SENAS). The SENAS has undergone extensive development as a battery of cognitive tests relevant to diseases of aging [71,72,73,74]. Episodic memory is a composite score derived from a multi-trial word-list-learning test (Word List Learning 1) [71]. Measure development and psychometric characteristics are described in more detail elsewhere [71,73,75]. Episodic memory scores are presented in z- score like units where a score of zero corresponds to the mean and differences from the mean are expressed in standard deviation units.

#### 3.2.4. Methods

We used a Mendelian randomization (MR) framework [76] to evaluate the potential role of reverse causation for the association between neighborhood NDVI and episodic memory. MR is a type of instrumental variable (IV) analysis that uses genetic variants as instruments or proxies for the exposure of interest, based on the idea that genes are natural experiments because they are randomly assorted and determined at birth [77,78]. APOE genotype is a strong genetic determinant of AD [79], which can be used to estimate the effect that cognition may have on neighborhood choice and characteristics (e.g., reverse causality). The presence of at least one APOE ε4 allele is associated with cognitive impairment in middle aged and older adults, particularly episodic memory [80], which is characteristically affected in Alzheimer’s disease (AD).

Genetic IV and MR studies allow for estimates of the causal effect of an exposure on outcome under a set of criteria outlined in Section 2.5 (i.e., instrument must be associated with outcome; instrument must affect outcome only through exposure, instrument not associated with confounders of exposure and outcome, estimated effects of instruments are the same across differing levels of instrument). However, even in the case that not all these criteria are met, MR can be used to identify shared etiologies (confounding) or reverse causation of a disease on an exposure.

We ran a series of models to evaluate the use of APOE genotype as an instrumental variable and compare observational and IV estimates for the effect of cognition on NDVI. All models were based on linear regressions and included robust standard errors (clustered by site) and controlled for age, sex, education, race/ethnicity, and recruitment source.

Our primary IV estimate used a two-stage least squares model (2SLS) [41], with APOE ε4 allele acting as an instrument for the association between episodic memory and NDVI. We examined the 1st stage regression results to confirm the presence of at least one APOE ε4 allele (versus none) was associated with episodic memory (F-statistic > 10 is suggested) [44]. This is an important requirement for suitability of APOE genotype as an instrumental variable (see Figure 2) [81]. We also ran a separate two-stage regression approach. First, we calculated values from the 1st stage regression of APOE ε4 allele to predict episodic memory. In the 2nd stage, we ran a linear regression using the predicted episodic memory values as the main exposure with NDVI as the outcome and calculated bootstrapped standard errors.

We also examined the linear regression estimate of episodic memory and NVDI (i.e., an observational estimate) adjusted for covariates to compare to our IV estimates.

#### 3.2.5. Results

Study participant characteristics are presented in Table 6. We confirmed that having at least one APOE ε4 allele was significantly associated with lower episodic memory score (*p* < 0.001, F-statistic= 45) in this sample and supports the relevance of APOE genotype as an instrument for episodic memory (Table 7). Linear regression (observational) and IV estimates (from 2nd stage models) all suggested an inverse effect of lower episodic memory on NDVI (Table 8). The IV estimates, using APOE ε4 allele as an instrument for episodic memory, had slightly stronger magnitude of the effect (−0.103 and −0.102) compared to the linear regression model (−0.013). The linear regression estimate can be interpreted as 1 SD higher episodic memory being associated with a 0.013 lower mean NDVI value for the neighborhood (95% CI: −0.018, −0.007). The IV estimates could be interpreted as the causal effect of 1 SD higher episodic memory score on NDVI (e.g., −0.103 [95% CI: −0.133, −0.074] for the 2SLS IV analysis), as predicted by APOE ε4 allele and covariates.

#### 3.2.6. Case Study 2 Conclusions

We used an IV/MR approach to examine the potential for reverse causation to explain some of the association between cognition and NE. Although this case study was in a small sample and was meant to be exploratory, we found inverse associations between episodic memory score and neighborhood mean NDVI. APOE ε4 allele was a strong predictor of episodic memory (1st stage regression), an important criterion for use as an instrumental variable for episodic memory. Using APOE ε4 allele as an instrument, estimates from our 2nd stage IV regressions suggested an effect of cognition on NDVI. Estimates were similar between two IV approaches and after adjustment for other potential factors that may lead to differences of APOE ε4 allele such as a race/ethnicity. The estimates for the IV were in the same direction of association as the observational estimates, but the magnitude of the effect was even stronger for the IV (e.g., −0.013 vs. −0.103). This suggests that the inverse association between cognition and NE in this setting may be driven by reverse causation, although other mechanisms or confounding in the opposite direction may also impact the observational estimates. We do not know the exact mechanisms through which individuals with declining memory or the APOE ε4 allele might prefer greener spaces. However, we might expect that individuals with declining memory may be more likely to move to greener places because this is where nursing homes and assisted living facilities tend to be located (suburbs). Together these preliminary findings suggest that reverse causation could be an issue in studies of NE and cognition in older adults, although future larger studies that have additional genetic information would be needed for verification. Longitudinal study designs of individuals in mid-life may be needed to help avoid substantial effects of reverse causation. Similar to the first case study, this second case study is exploratory and aimed at demonstrating a method that could be used to examine reverse causation in future NE and brain health studies. Thus, estimates may not reflect a true causal effect of episodic memory on neighborhood NDVI and should be interpreted cautiously.

Employing this IV/MR approach may help estimate the contribution of reverse causation which could be used to adjust observational estimates for the effect of cognition on NE. Additionally, other IV and natural experiments for NEs may help provide evidence on the role of NE on cognitive functioning.

## 4. Conclusions

In this paper, we outlined the most common methods used and proposed to address self-selection and reverse causation in studies of NEs and health. We discussed the strengths and pitfalls of using these methods when they are extended to studies of brain health and ADRD, and we provide a qualitative summary of the possibility for each method to successfully account for self-selection and reverse causation and thus improve causal inference (Table 2). Cross-sectional analyses of NEs and brain health remain more common than longitudinal or quasi-experimental studies and have a high chance of bias without careful consideration of self-selection and reverse causality, as well as the other unique methodological considerations that must be weighed when studying neighborhood characteristics and brain health outcomes (Table 1).

Our case studies demonstrated four of the methods outlined in this paper, namely: (1) adjustment for self-selection as a covariate; (2) restriction/stratification by the self-selection variable; (3) inverse probability weighting (from propensity scores) based on the probability of living in a neighborhood; and (4) instrumental variable analysis to investigate the potential for reverse causation (e.g., brain health leads to the neighborhood characteristic). The results from our case studies reiterate that methodological choices can have an impact on study findings and that studies of NEs and brain health would ideally conduct sensitivity analyses, whenever possible, to determine if results hold using different methods. Consistent findings across the methods may provide support for a true association, whereas inconsistent findings may help explain potential differences in associations observed between similar studies previously published. Authors should openly discuss their data and methodological limitations. If data or resources are not available to employ more rigorous methods, the improved/aspirational methods should also be acknowledged in the limitations sections of papers.

Few studies of NE and brain health have yet to incorporate the methods discussed in this review. Thus, we have provided more established methods used in public health and social science fields that can assist in addressing causality when confronted with potential self-selection or reverse causation. Methods borrowed from related disciplines (e.g., geography, statistics, or highly specialized medical fields) may further improve our ability to establish causality and should be investigated in future methodological studies. In addition, a mixed methods approach [82], where researchers use both quantitative and qualitative approaches (e.g., focus groups, interviews), may be fruitful in capturing key information to elucidate the causal timing of exposure and outcome (i.e., addressing reverse causality) and neighborhood preference and residential moves due to factors related to brain health (i.e., addressing self-selection).

The next wave of NE and brain health papers need to advance beyond the traditional cross-sectional and longitudinal studies designs with multivariable regression analyses to strengthen causal inference. Changes to NEs can be costly, time consuming, and it can be difficult to garner political and public support. Proposed changes to NEs based on methodologically unsound health studies could result in unintended harms to population health that may be avoidable with more rigorously conducted studies. Neighborhood interventions, policies, and programs to improve health behaviors, exposures, and reduce the risk of ADRD will be best supported by minimally biased studies that have a critical eye on causal inference.

## Figures and Tables

**Figure 1 ijerph-18-06484-f001:**
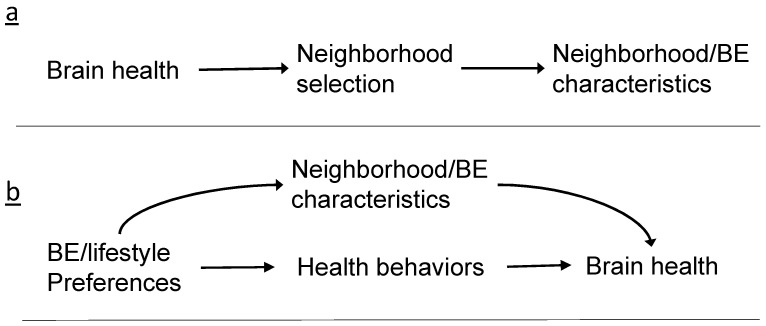
Illustration of relationship between neighborhood/built environment (BE) and brain health in the case of bias by (**a**) reverse causation; (**b**) self-selection by individual preferences.

**Figure 2 ijerph-18-06484-f002:**
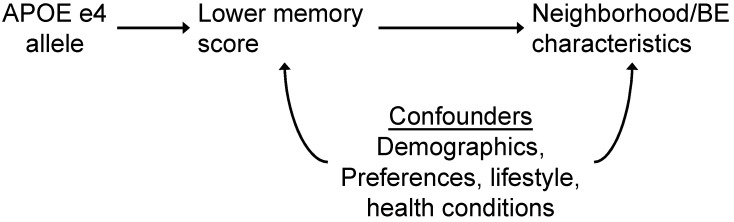
APOE genotype as instrumental variable to study potential reverse causality of association between neighborhood NDVI and episodic memory. If APOE is associated with NE/BE characteristics, it is likely through cognition and provides support for reverse causality.

**Table 1 ijerph-18-06484-t001:** Select methodological challenges of research on neighborhood environments and brain health.

Challenge	Example Issues to Consider
Defining neighborhood boundaries: self-reported/perceived, administrative boundaries (e.g., US Census tract), or geographic information system (GIS) buffers around the residence (e.g., ½-mile Euclidian buffer)	Self-reported measures: Difficult to compare self-reported measures across participants. Administrative boundaries: Varying land area, lack of international comparability, may not represent neighborhood area most pertinent to individual. GIS buffers or administrative boundaries: May not represent neighborhood area most pertinent to individual.
Capturing neighborhood exposure: time period, place, degree	What time period of exposure is most important for brain health (childhood, middle age, late life)? Where is exposure most pertinent and does this depend on life stage? How do we quantify degree of exposure? Should we/how to consider accumulated exposure?
Defining the neighborhood construct	Can we develop neighborhood measures that have high validity and reliability?
Self-selection into neighborhoods	Individuals may choose to move to neighborhoods because they offer opportunities for health behaviors (e.g., walking, healthy foods) that affect brain health.
Reverse causation: Association is due to outcome leading to exposure, not vice versa	Alzheimer’s disease and related disorders lead to neighborhood selection (e.g., brain health outcome is related to residential move in late life to accommodate health needs).
Strong correlation of neighborhood characteristics	When highly correlated, how do we know the association found for one variable is not actually demonstrating effect of highly correlated variable?
Neighborhood segregation	Structural racism/exclusionary and discriminatory policies and practices led to residential segregation of racial/ethnic groups and socioeconomic status that is highly correlated and difficult to disentangle from other neighborhood characteristics (e.g., access to parks and healthy foods) [23,24,25].
Spatial considerations	Residential areas that are closer together tend to have similar values (e.g., similar exposures and/or outcomes), which if not accounted for in the analyses can lead to erroneous conclusions. Modifiable area unit problem: the area unit employed to define the neighborhood (e.g., Census tract versus Census block group) can affect the significance of findings.
Studying older adults (e.g., >60 years old)	Older adults are more likely to develop physical and cognitive impairments that can affect study enrollment, attrition, and participation in study procedures (e.g., magnetic resonance imaging). Requiring survival to old age may result in highly select samples who are healthier.
Lag between Alzheimer’s disease and related disorders pathology development and dementia diagnosis	Longitudinal studies of older adults may not sufficiently account for undiagnosed, prodromal disease affecting neighborhood exposure.
Invasive/time consuming procedures to measure brain health (e.g., lumbar puncture, brain imaging) may limit the types of neighborhoods or ranges of neighborhood characteristics captured	Restricting to individuals who consent to/complete invasive brain health procedures are more likely to include individuals of White race and higher socioeconomic status, who typically live in White neighborhoods with higher socioeconomic status. This can limit generalizability and result in selection bias.

**Table 2 ijerph-18-06484-t002:** Methods to address self-selection and reverse causality in neighborhood environment and brain health studies.

	Potential to Address:	
Method	Neighborhood Self-Selection	Reverse Causality
Randomized control trial	++++	++++
Multivariable regression: covariate adjustment for self-selection	+	
Multivariable regression: propensity score matching, inverse probability weighting	++	
Longitudinal study design	++	++
Restriction/stratification of sample	+	+
Quasi-experiment: natural experiment, instrumental variable analysis	+++	+++

Abbreviations: Qualitative scoring: no + = no potential; + low potential; ++ moderate potential; +++ moderate to high potential; ++++ High potential.

**Table 3 ijerph-18-06484-t003:** Case Study 1 Participant Characteristics.

Characteristic	
Age, n (%)	
18–49 years	49 (32.5%)
50–64 years	58 (38.4%)
65 and older	44 (29.2%)
Women, n (%)	91 (60.2%)
Married/with partner, n (%)	105 (69.5%)
Race, n (%)	
White	128 (84.8%)
Other	23 (15.2%)
Annual family income, n (%)	
<$50,000	15 (9.9%)
$50,000–99,999	41 (27.2%)
$100,000–$149,999	42 (27.8%)
≥$150,000	53 (35.1%)
Employed, n (%)	104 (68.9%)
Children living in household, n (%)	50 (33.1%)
Always drive places (yes, self-report), n (%)	72 (47.7%)
Accessible neighborhood parks/open space (yes, self-report), n (%)	108 (71.5%)
Ability to walk to shops and dining (yes, self-report), n (%)	65 (43.1%)
Minutes walking/week, mean (SD)	131.2 (303.5)
Minutes bicycling/week, mean (SD)	37.3 (79.4)

Abbrevations: SD, standard deviation.

**Table 4 ijerph-18-06484-t004:** Association between living in Abacoa neighborhood and minutes walking and bicycling per week, controlling for neighborhood self-selection as covariate versus restricting by the neighborhood self-selection variable.

Outcome	Adjusting for Self-Selection of Neighborhood Amenities as Covariate (Model 1) N = 151	Restricted to Those Reporting Neighborhood Amenities Were:	
		Primary Reason for Choosing Home (Model 2) n = 57	Not primary Reason for Choosing Home (Model 3) n = 94
	Estimate (95% CI)	Estimate (95% CI)	Estimate (95% CI)
Minutes of neighborhood walking/week	**128.8 (15.5, 242.1)**	227.2 (−65.4, 419.8)	55.2 (−1.71, 112.2)
Minutes of bicycling/week	**31.6 (2.6, 60.7)**	61.5 (−2.5, 125.5)	19.5 (−11.6, 50.6)

All linear regression models (n = 151) controlled for age, sex, education, income, race/ethnicity, children living in household, married/with partner, employed, always drive places, appraisal of neighborhood availability of parks/open space and ability to walk to shops/dining; Model 1 additionally controlled for self-selection of home due to nearby neighborhood amenities. Bold = statistically significant at *p* < 0.05.

**Table 5 ijerph-18-06484-t005:** Association between living in Abacoa neighborhood and minutes walking and bicycling per week, using and not using inverse probability weights to account for neighborhood self-selection.

Outcome	Model 1—with no IPW	Model 2—with IPW
	Estimate (95% CI)	Estimate (95% CI)
Minutes of neighborhood walking/week	**132.1 (19.4, 244.8)**	**96.7 (2.1, 191.3)**
Minutes of bicycling/week	**33.8 (4.6, 63.0)**	24.2 (−2.6, 51.0)

Abbreviation: IPW = inverse probability weighting; Linear regression models (n = 151) controlled for age, sex, education, income, race/ethnicity, children living in household, married/with partner, employed, always drive places, appraisal of neighborhood availability of parks/open space and ability to walk to shops/dining; Model 2 was weighted by inverse probability weights for probability of living in Abacoa versus the other surveyed South Florida neighborhoods. Bold = statistically significant at *p* < 0.05.

**Table 6 ijerph-18-06484-t006:** Case Study 2 Participant Characteristics.

Characteristic	Statistic
Total sample, N	243
Age, mean (SD)	76.0 (7.1)
NDVI, mean (SD)	−0.08 (0.09)
Episodic memory, mean (SD)	−0.35 (0.86)
Female, n (%)	140 (57.6)
Race/ethnicity, n (%)	
Black, non-Hispanic	62 (25.5)
White, non-Hispanic	111 (45.7)
Hispanic	55 (22.6)
Other	15 (6.2)
APOE ε4 allele carrier, n (%)	107 (44.0)
Community-based recruitment (vs clinic), n (%)	190 (78.2)
Bay Area site (vs Sacramento), n (%)	120 (49.4)

Abbreviation: NDVI, Normalized Difference Vegetation Index; SD, standard deviation.

**Table 7 ijerph-18-06484-t007:** 1st Stage regression results for strength of APOE genotype as an instrument for episodic memory.

	Episodic Memory		
	Estimate (95% CI)	*p*-Value	F Statistic
APOE ε4 allele	−0.22 (−0.29, −0.17)	<0.001	45.4

Based on linear regression model with robust standard errors for site and controlling for age, sex, education, race/ethnicity, recruitment source.

**Table 8 ijerph-18-06484-t008:** Linear regression and instrumental variable (IV) estimates for the effect of episodic memory on neighborhood Normalized Difference Vegetation Index.

Model	NDVI
	Estimate (95% CI)
Linear regression (observational)	−0.013 (−0.018, −0.007)
2 stage least square IV	−0.103 (−0.133, −0.074)
Separate 2 stage IV, bootstrapped standard errors	−0.102 (−0.205, −0.008)

Abbreviation: IV = instrumental variable; NDVI = Normalized Difference Vegetation Index; Models included clustering by site and controlled for age, sex, education, income, race/ethnicity, recruitment.

## Data Availability

Data inquiries regarding NDVI can be made to lbesser@fau.edu. Requests for UC Davis ADRC data can be made to lwjin@ucdavis.edu.

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
