# Peer review of "Methods to Address Self-Selection and Reverse Causation in Studies of Neighborhood Environments and Brain Health"

_ijerph, 2021, doi:10.3390/ijerph18126484_

Round 1

Reviewer 1 Report

This study evaluates the ways in which Alzheimer’s Disease and Related Disorders (ADRD) have typically been measured in older adults in research focusing on neighborhood effects. The authors review current methods, conduct case studies of two potential methods, and provide recommendations on how to proceed with this research in a way that improves on typical approaches. The manuscript is very well-written, the arguments clear and compelling, and the case studies offer insight into some directions forward.

To that point, I have no line editing or specific changes to recommend; however, there are some conceptual issues that I think deserve greater attention in this study.

First, the paper tends to assume that current approaches are a good starting place from which to improve, and the recommendations made for future studies are essentially iterations of current methods. I would instead challenge the authors to ask themselves whether all of these approaches are valid on their face, e.g., is it really appropriate to assume we can boil down complex BE-health issues to an instrumental variable approach? I’m not sure it is.

Another example – Moving to Opportunity is the only study of its kind (and did not focus on older adults) for a reason, which the authors did well to explain, i.e., it’s nearly impossible to do RCTs in environmental context, and again I would ask, is that really a deficit? Or should we perhaps be more creative in thinking about neighborhood-level approaches and not be so willing to rely on adapted clinical methods?

As to coming to agreement on neighborhood definition, I’m also not convinced this is something for which we should be aiming. There’s a case to be made that peoples’ perceptions of “your neighborhood” may be more important for quantifying the impacts of BE characteristics on their health outcomes than any contrived definition, and I would think this may be an especially important point to consider with people facing ADRD.

The issue noted with regard to disentangling residential segregation from confounders is well-considered. There is a growing literature directly linking the physically identifiable legacies of redlining and racial zoning to health outcomes in the United States, and I suggest noting this is a promising approach, especially if there are similar historical boundary patterns that can be identified in other countries as well. References for example.1-3

Finally, there’s a distinct lack of recognition among the methods reviewed regarding spatial techniques and the contributions of geographers, as well as mixed methods approaches that are better equipped to ascertain the aspects of “neighborhood” that residents perceive to be impacting them on a daily basis. See references for examples.4-8

A major deficit of this literature is the inattention to spatial variation regarding what are inherently spatial units, e.g., neighborhoods, as well as what place-making means for residents, especially those increasingly in need of familiar landmarks to help them navigate their environments as their cognition begins to decline.

My anticipation is not that the authors should completely revamp their study, but I do suggest engagement with the geographic and mixed methods literature in the Discussion and broader acknowledgment that there are other ways forward that perhaps need additional assessment, but could also bring a new light to bear on the topic at hand.

  1. Nardone A, Chiang J, Corburn J. Historic Redlining and Urban Health Today in U.S. Cities. Environmental Justice. 2020;13(4):109-119.
  2. Krieger N, Wright E, Chen JT, Waterman PD, Huntley ER, Arcaya M. Cancer Stage at Diagnosis, Historical Redlining, and Current Neighborhood Characteristics: Breast, Cervical, Lung, and Colorectal Cancers, Massachusetts, 2001–2015. American Journal of Epidemiology. 2020;189(10):1065-1075.
  3. Krieger N, Wye GV, Huynh M, et al. Structural Racism, Historical Redlining, and Risk of Preterm Birth in New York City, 2013–2017. American Journal of Public Health. 2020;110(7):1046-1053.
  4. McQuoid J, Thrul J, Ling P. A geographically explicit ecological momentary assessment (GEMA) mixed method for understanding substance use. Social Science & Medicine. 2018;202:89-98.
  5. Curtis A, Curtis JW, Ajayakumar J, Jefferis E, Mitchell S. Same space – different perspectives: comparative analysis of geographic context through sketch maps and spatial video geonarratives. International Journal of Geographical Information Science. 2019;33(6):1224-1250.
  6. Pearce JR, Richardson EA, Mitchell RJ, Shortt NK. Environmental justice and health: the implications of the socio-spatial distribution of multiple environmental deprivation for health inequalities in the United Kingdom. Transactions of the Institute of British Geographers. 2010;35(4):522-539.
  7. Kolak M, Bhatt J, Park YH, Padrón NA, Molefe A. Quantification of Neighborhood-Level Social Determinants of Health in the Continental United States. JAMA Network Open. 2020;3(1):e1919928-e1919928.
  8. Root ED, Silbernagel K, Litt JS. Unpacking healthy landscapes: Empirical assessment of neighborhood aesthetic ratings in an urban setting. Landscape and Urban Planning. 2017;168:38-47.

Reviewer 2 Report

To the authors of “Methods to address self-selection and reverse causation in studies of
neighborhood environments and brain health:”
This study aims to offer a review of the literature related to selection bias and reverse causation
related to neighborhood effects. The neighborhood effects discussed in the paper are whether
people affected by dementia and Alzheimer select specific neighborhoods because of their
characteristics and the extent to which such neighborhoods characteristics have an independent
effect of cognitive health. The authors also offer their own analysis along two case studies.
The paper is very well written and articulated, and I think that it will make a significant contribution
to the literature once it is revised.
In my opinion, there are three issues that need to be addressed.
1. The review of methods addressing selection bias needs to be amended by including the
difference-in-difference specification. Many studies focusing on neighborhood effects estimate a
multivariable regression model with a difference-in-differences specification. I would recommend
that the authors include a review of these studies in the paper.
2. My second concern is that there is not enough detail in the first case study. I would like to see
proper tables with all characteristics used in the study, with the number of observations. Also, the
regression analyses need to be reported in respective tables, again with all appropriate details.
3. My third concern is about the use of an instrumental variable in the second case study. As Varian
argues (2016), some potentially observable part of A (the treatment, in this case episodic memory) is
independent of the error term and thus allows us to see how an essentially random variation in A
affects the outcomes, in this case NDVI. “A variable that affects Y only via its effect on A is called
an instrumental variable,”
?? → ? → ?
Following Varian’s model specification recommendation (also Nguyen et al., 2016), at the first-stage
linear regression the authors need to predict A (episodic memory) based on APOE 4 allele and then
use the predicted values in the second-stage regression, the model specification can vary. The causal
estimate is this second-stage regression coefficient, which expresses the change in NVDI caused by
a unit change in the predicted A, from the first-stage regression (Varian, 2016; Burgess, Small and
Thompson, 2017).
I would like to see a clearer explanation of these effects (including tables with the full two-stage
regression results) before I can properly judge the conclusions from this analysis. Below I am pasting
the references and suggest that the authors check these out:
Varian, Hal. R. 2016. “Causal Inference in Economics and Marketing.” PNAS4" | 2016/3/29 |
15:25.
Burgess, Stephen, Dylan S Small and Simon G Thompson. 2017. “A review of instrumental variable
estimators for Mendelian randomization.” Statistical Methods in Medical Research, 2017, Vol. 26(5)
2333–2355.2
Nguyen, Thu T., ScD, MSPH, et. al. 2016. “Instrumental variable approaches to identifying the
causal effect of educational attainment on dementia risk.” Annals of Epidemiology 26 (2016) 71e76

Round 2

Reviewer 2 Report

To the authors of “Methods to address self-selection and reverse causation in studies of neighborhood environments and brain health:” 

Thank you for taking into consideration mine and the other reviewers’ recommendations, I think that the paper is much stronger for it. I have two minor additional comments. 

In the first case study, the number of non-White respondents is quite low – only 15 percent and it is also a composite category, meaning that there is wide variation within the category. We don’t really know if most of these respondents are Latinx, African American, Asian, etc. Such within category variation is associated with inflated standard errors and therefore biased estimates. Particularly since residential preferences vary significantly by racial/ethnic group. I would recommend that you include a cautionary note about the possible bias in the results related to the effects of race/ethnicity while modeling investigating self-selection. 

The second case study has a much better racial/ethnic composition, it looks much improved and again I appreciate the thoughtful revision of this part. I am a little worried about the small number of observations and am happy to see you pointing out that this is more of an exploratory study than a true casual inference one. That should be emphasized in the paper. The IV identification strategy is one of the most difficult in the scholarly literature (and among the most controversial) but I like the attempt to use it in modeling relationships. My second concern is that I am not a neurologist and expert in the links between genetic information and episodic memory, therefore I trust that the selection of this particular IV makes sense from a medical standpoint. Hopefully some of the other reviewers have better expertise in that context than me. 

All in all, very thorough revision and excellent paper! 
